# Micro-RNAs in Response to Active Forms of Vitamin D_3_ in Human Leukemia and Lymphoma Cells

**DOI:** 10.3390/ijms23095019

**Published:** 2022-04-30

**Authors:** Justyna Joanna Gleba, Dagmara Kłopotowska, Joanna Banach, Karolina Anna Mielko, Eliza Turlej, Magdalena Maciejewska, Andrzej Kutner, Joanna Wietrzyk

**Affiliations:** 1Department of Experimental Oncology, Hirszfeld Institute of Immunology and Experimental Therapy, Polish Academy of Sciences, Weigla 12, 53-114 Wroclaw, Poland; dagmara.klopotowska@hirszfeld.pl (D.K.); joanna.banach@hirszfeld.pl (J.B.); karolina.mielko@pwr.edu.pl (K.A.M.); eliza.turlej@upwr.edu.pl (E.T.); magdalena.maciejewska@hirszfeld.pl (M.M.); joanna.wietrzyk@hirszfeld.pl (J.W.); 2Department of Biochemistry, Molecular Biology and Biotechnology, Faculty of Chemistry, Wroclaw University of Science and Technology, Norwida 4/6, 50-373 Wroclaw, Poland; 3Department of Experimental Biology, The Wroclaw University of Environmental and Life Sciences, Norwida 27 B, 50-375 Wroclaw, Poland; 4Department of Bioanalysis and Drug Analysis, Faculty of Pharmacy, Medical University of Warsaw, 1 Banacha, 02-097 Warsaw, Poland; andrzej.kutner@wum.edu.pl

**Keywords:** leukemia, lymphoma, miRNAs, vitamin D_3_ analogs, calcitriol, tacalcitol, cell cycle, aneuploids, miR-27b, miR-125b

## Abstract

Non-coding micro-RNA (miRNAs) regulate the protein expression responsible for cell growth and proliferation. miRNAs also play a role in a cancer cells’ response to drug treatment. Knowing that leukemia and lymphoma cells show different responses to active forms of vitamin D_3_, we decided to investigate the role of selected miRNA molecules and regulated proteins, analyzing if there is a correlation between the selected miRNAs and regulated proteins in response to two active forms of vitamin D_3_, calcitriol and tacalcitol. A total of nine human cell lines were analyzed: five leukemias: MV-4-1, Thp-1, HL-60, K562, and KG-1; and four lymphomas: Raji, Daudi, Jurkat, and U2932. We selected five miRNA molecules—miR-27b, miR-32, miR-125b, miR-181a, and miR-181b—and the proteins regulated by these molecules, namely, CYP24A1, Bak1, Bim, p21, p27, p53, and NF-kB. The results showed that the level of selected miRNAs correlates with the level of proteins, especially p27, Bak1, NFκB, and CYP24A1, and miR-27b and miR-125b could be responsible for the anticancer activity of active forms of vitamin D_3_ in human leukemia and lymphoma.

## 1. Introduction

Disorders of crucial stages of hematopoiesis, such as self-renewal, proliferation, and differentiation, are the leading causes of hematopoietic and lymphatic neoplasms. An essential element is unblocking the differentiation process and directing the cells to apoptosis. Therapies lead to complete remission and thus extend the patient’s survival time. One of the therapies that offer such a possibility is differential therapy using a retinoic acid derivative (ATRA) [1]. The use of ATRA was initiated to treat patients with acute promyelocytic leukemia, achieving complete disease remission. Thanks to differential therapy, this fatal disease became utterly curable. Further clinical trials confirmed the high effectiveness of ATRA, also in combination with conventional chemotherapy [2,3,4]. The appearance of the so-called differentiation syndrome (DS), i.e., the retinoic acid syndrome, is a severe and prevalent complication after such treatment [5]. Therefore, it constantly emphasizes developing new, combined differential therapies that target specific abnormalities underlying cancer pathogenesis. Attempts are being made to use the active form of vitamin D_3_—calcitriol—and its derivatives in patients with myeloid neoplasms [6,7]. Oncogenic microRNAs play a key role in the development of leukemia and lymphoma [8,9]. Moreover, they could be biomarkers and molecular factors in diagnosing and treating leukemias and lymphomas [10,11,12]. It turns out that miRNA molecules can directly regulate the actions of active forms of vitamin D_3_. In the 3 ‘UTR region of the classic vitamin D receptor (VDR) mRNA, there is an element recognized by miR-125b [12]. Studies have shown that its overexpression reduces the level of vitamin D receptor protein in MCF-7 breast cancer cells by 40% compared to the control [12]. miR-125b is not the only molecule that regulates the level of the vitamin D receptor, as miR-27b also binds at the 3 ‘UTR site of the gene for the classic vitamin D receptor, reducing its expression, which was observed in cancer cells of the pancreas and colon [13]. Similarly, the expression of the CYP24A1 enzyme, which converts calcitriol, is regulated by miR-125b, suggesting that the level of this enzyme may be correlated with the level of miR-125b expression [14]. The miR-125b molecule, in addition to regulating the expression of proteins directly responsible for the action of calcitriol, also controls other key proteins of the cell [15,16]. The participation of this molecule in many essential processes, e.g., in apoptosis, suggests that it could become a target in anti-cancer therapy [17,18]. Other reports indicate that VDR may upregulate the expressions of miR-125b and miR-27b. This shows a more complex mechanism of interaction of these miRNAs and vitamin D_3_ [19,20,21]. miR-125b regulates apoptosis in mature hematopoietic cells by blocking the pro-apoptotic Bak1 protein, is also responsible for the resistance to chemotherapeutic agents used to treat leukemia, such as vincristine and daunorubicin, and plays a significant role in the regulation of p53 protein expression and influences the activity of the NF-κB factor [22,23,24]. It is known that the use of calcitriol may affect the expression profile of miRNA molecules in cells. Experiments on the HL-60 acute promyelocytic leukemia and U937 lymphoma cells proved that calcitriol reduced the expression of miR-181a and miR-181b [25]. Decreasing the level of these non-coding structures increased the level of mRNA and p27 protein [26]. The p27 protein is directly related to monocyte differentiation [27]. Calcitriol also decreases miRNAs such as miR-302c and miR-520c in Kasumi-1 and K562 human leukemia cells, increasing their sensitivity to NK cell activity [28]. The expression of miR-32 is also reduced after the use of calcitriol, which is associated with an increase in the level of the pro-apoptotic protein Bim and sensitizes AML cells to the cytostatic cytosine arabinoside [29]. Similarly, the effect of calcitriol was noticed against miR-26a in HL-60 acute promyelocytic leukemia cells, which was associated with the regulation of factor E2F7, which causes an increase in p21 protein expression [30]. Moreover, calcitriol caused a significant reduction in the levels of miR-17-5p/20a/106a, miR-125b, and miR-155, which are repressors for AML1, VDR, and C/EBPβ in acute promyelocytic leukemia HL-60 cells [31]. In the conducted research, we analyzed the level of the miRNA molecules and the regulated proteins in human leukemia and lymphoma cells. The level of tested molecules was analyzed in response to calcitriol and tacalcitol.

## 2. Results

### 2.1. Calcitriol and Tacalcitol Effect on the Cell Cycle in Human Leukemia and Lymphoma

Calcitriol and tacalcitol increased the percentage of cells in the G_1_/G_0_ phase of the cell cycle. The changes were observed only in the MV-4-11, Thp-1, and HL-60 cell lines. These cell lines have been defined as sensitive to calcitriol and tacalcitol. In the case of the MV-4-11 cell line, an increase in cells in the G_1_/G_0_ phase was observed after 48 h, and this effect lingered until 120 h after calcitriol and tacalcitol use. At the same time, the population of cells in the S phase of the cell cycle decreased. The most significant reduction in the percentage of cells in the S phase was observed after 72 h, as compared to the control cells. There were no significant changes in the percentage of cells in the G_2_/M phase of the cell cycle (Figure 1a). A similar effect of an increase in the percentage of cells in the G_1_/G_0_ phase of the cell cycle was observed in the Thp-1 cell line. Ninety-six hours after using the tested compounds, an increase in the percentage of cells in the G_1_/G_0_ phase by about 20% compared to control cells was observed, with a simultaneous decrease in the number of cells in the S phase of the cell cycle. However, there were no significant changes in the percentage of cells in the G_2_/M phase of the cell cycle (Figure 1b).

The use of calcitriol and tacalcitol showed a similar effect against the third sensitive cell line HL-60. Incubation with the tested compounds increased the percentage of cells in the G_1_/G_0_ phase. A slight effect of the substances used was observed after 24 h of incubation, but the most remarkable differences were noted after 72 h. Around a 20% increase in the percentage of cells in the G_1_/G_0_ phase was observed compared to the control cells. In addition, the population of cells in the S phase of the cell cycle was reduced. There were no changes in the percentage of cells in the G_2_/M phase (Figure 1c). After calcitriol and tacalcitol use, there were no statistically significant changes in the other tested leukemia and lymphoma cells (Appendix A, Appendix A).

### 2.2. Population of Aneuploid Cells and Response to Calcitriol and Tacalcitol

The population of aneuploid cells was analyzed in human leukemia and lymphoma cells. Studies have shown that the highest percentage of aneuploid cells was present in the two sensitive to calcitriol and tacalcitol cell lines: MV-4-11 and Thp-1. About 14% of aneuploid cells were observed in the MV-4-11 cell line, while about 17% were observed in the Thp-1 cell line. Interestingly, the remaining leukemia cell lines were 10% aneuploids (Figure 2a). However, analysis of the lymphoma cells showed a low percentage of aneuploid cells. In Raji, about 2% of aneuploid cells were observed. Around 3% of Daudi cells were aneuploid, and only 1% in Jurkat and U2932 (Figure 2b).

Interestingly, the use of calcitriol and tacalcitol reduced the number of aneuploid cells in three sensitive cell lines: MV-4-11, Thp-1, and HL-60. Incubation of the MV-4-11 cells with calcitriol and tacalcitol reduced the number of aneuploids. The most significant effect of the tested substances was observed after 72 h when there was an approximately 10% decrease in the population of aneuploid cells. Lowering the number of aneuploid cells was observed after 96 and 120 h (Figure 3a). A similar effect was observed in Thp-1 cells. A decrease in the percentage of aneuploids was observed after 24 h, while significant differences were observed after 48 h of incubation with the tested compounds. Tacalcitol reduced the number of aneuploids by about 10% and calcitriol by about 5% (Figure 3b). A downward trend in the aneuploid population was also observed in HL-60 cells. A slight decrease in the percentage of aneuploids was observed 48 h after the treatment. However, a significant reduction occurred after 72 h, and this effect continued after 96 and 120 h. Interestingly, in the case of HL-60 cells, the percentage of aneuploids increased over time in the cell culture, and after 120 h, it was the highest, at about 15% (Figure 3c).

In the case of two other leukemia cell lines of KG-1 and K562 and all lymphoma cells, no changes in the percentage of aneuploid cells were observed after the calcitriol and tacalcitol treatment (Appendix A).

### 2.3. miR-181b and p27 Protein Involved in Cell Cycle

The basic level of p27 protein was significantly higher in the Raji and Daudi cells than PBMCs. However, the level of remaining cells was comparable to the control cells (Figure 4a). The use of calcitriol and tacalcitol resulted in an increased level of p27 in MV-4-11, HL-60, and Thp-1 cells. Conversely, calcitriol significantly reduced p27 in Raji and tacalcitol in KG-1, Daudi, and Jurkat cells (Figure 4b). miR-181a can impact the level of p21 protein by regulating the p53 protein, while miR-181b directly regulates the level of p27 [32,33]. It turned out that most cell lines have similar levels of the miR-181b molecule compared to PBMC. The exception was the HL-60 cell line, which had a much higher miR-181b than the control cells (Figure 4c). Interestingly, miR-181b decreased after using calcitriol and tacalcitol in three sensitive cell lines, MV-4-11, Thp-1, and HL-60. However, it increased in K562, Raji, Daudi, and Jurkat cells (Figure 4d). The miR-181a and p21 protein analysis did not show any correlation between the basal level and after treatment (Appendix A).

### 2.4. miR-32b and Pro-Apoptotic Bim Protein

The miR-32 molecule and the Bim protein levels were determined in human leukemia and lymphoma cells. The miR-32 level in all tested cell lines was significantly lower than in the PBMCs used as the control. In all leukemia cell lines, the level of Bim protein was significantly higher than the level observed in the control cells. Lymphoma cells had a similar level of Bim protein to the normal blood cells (Figure 5a). Cell incubation with calcitriol or tacalcitol caused a decrease in the Bim level in three sensitive cell lines: MV-4-11, Thp-1, HL-60, KG-1, Daudi, and Jurkat (Figure 5b). The analyzed level of the miR-32 molecule showed that all the tested cell lines had significantly lower levels compared to the control (Figure 5c). Interestingly, the active forms of vitamin D_3_ increased the level of miR-32 in three sensitive cell lines: MV-4-11, Thp-1, HL-60, K562, and Daudi. Calcitriol caused an increase in Jurkat, while tacalcitol decreased the level in this cell line (Figure 5d).

### 2.5. miR-125b and Regulated Proteins: Pro-Apoptotic Protein Bak1, NFκB, and p53, and CYP24A1

#### 2.5.1. miR-125b

The miR-125b in the tested cell lines showed that six cell lines: MV-4-11, Thp-1, K562, Raji, Daudi, and Jurkat, had a lower level of this molecule compared to the PBMCs. HL-60 had a similar level to the control cells, and two cell lines—KG-1 and U2932—had levels higher than the PBMCs (Figure 6a). The use of calcitriol and tacalcitol caused the increase of miR-125b in seven tested cell lines. No changes were observed in KG-1 and Raji. No significant differences also were observed after calcitriol use in the Daudi and U2932 cell lines (Figure 6b).

#### 2.5.2. Bak1

It turned out that the level of Bak1 protein in all myeloid neoplasms and Daudi and Jurkat cells was similar to the level present in PBMCs. A significantly higher pro-apoptotic protein Bak1 than in the control cells was observed in Raji and U2932 lymphoma cells (Figure 7a). Calcitriol and tacalcitol caused significant upregulation of the Bak1 protein in the sensitive cells MV-4-11, Thp-1, and HL-60. Interestingly, calcitriol itself caused a reduction in the level of Bak1 protein in K562 and Daudi cells (Figure 7b).

#### 2.5.3. NFκB Protein

A protein that the miR-125b molecule regulates is also the transcription factor NFκB. The analysis of the level of this molecule in cells not treated with calcitriol and tacalcitol showed that MV-4-11 and Daudi cells had levels similar to those observed in normal cells. On the other hand, Thp-1, Jurkat, and U2932 cells have much more significant NFκB protein levels than normal cells. The remaining cells of the HL-60, K562, KG-1, and Raji lines selected for the study had lower levels compared to the control cells (Figure 8a). After calcitriol and tacalcitol treatment, the level of NFκB was downregulated in three sensitive cell lines: MV-4-11, Thp-1, and HL-60. There were no significant changes in the other cell lines (Figure 8b).

#### 2.5.4. CYP24A1

Analysis of the amount of protein in cells not treated with calcitriol and tacalcitol showed that all the cell lines tested had a significantly higher level of the CYP24A1 compared to the control PBMCs (Figure 9a). However, only in the sensitive cells, namely, MV-4-11, Thp-1, and HL-60, an increase in the level of the CYP24A1 protein was observed after the use of calcitriol and tacalcitol (Figure 9b).

### 2.6. miR-27b

These studies analyzed two microRNA molecules that regulate the classical vitamin D receptor levels: miR-27b (Figure 10) and miR-125b (Figure 6). The level of the miR-27b molecule was lower in all tested cell lines compared to the PBMCs (Figure 10a). Cell incubation with calcitriol and tacalcitol increased miR-27b in almost all tested cell lines. The K562 cell line was the only cell line where no significant changes were observed. In addition, there were no statistically significant changes in Thp-1, KG-1, Raji, and Daudi after using calcitriol (Figure 10b).

### 2.7. Selected mRNA and miRNA Expression Level and Overall Survival

Using The Cancer Genome Atlas (TCGA) data, we correlated the expression level of selected mRNAs and miRNAs with overall survival (OS). The analyses were performed for acute myeloid leukemia. No data on the expression of the studied genes in patients with lymphoma were found. The analysis showed that the high level of the five examined genes are a poor prognosis factor in AML. These genes included VDR, Bak1, Bim, p21, and NFκB. Interestingly, the high expression level of p27 and p53 and all selected miRNAs was a good prognostic marker, extending the life span for patients with AML (Figure 11). There was no data about the CYP24A1 expression level and OS in AML patients.

## 3. Discussion

Accumulating evidence indicates that small, non-coding microRNAs (miRNAs) are involved in the cellular response to calcitriol [34]. MicroRNAs are RNA molecules (usually 18 to 25 nucleotides) that play a regulatory role in plant and animal cells. They attach to the messenger RNA of target genes and control their expression and basic cell processes. Mature miRNA molecules anneal complementarily to the 3 ‘UTR region of the target gene mRNA, blocking and reducing its expression [35,36,37]. Using computational predictions, studies published by Calin et al. showed that one cluster containing miR-15a/16-1 holds about 14% of the genes in the human genome [38]. Studies confirm that microRNAs can directly regulate tumor growth. In 2006, Costinean et al., using transgenic mice overexpressing miR-155, proved that this molecule is responsible for developing B-cell neoplasms [39]. This experiment shows that miRNAs are essential in understanding the complex process of cancer formation. Many studies confirm that changes in the miRNA expression profile correlate with cancer development. Moreover, numerous experiments show that these molecules are responsible for cancer cells’ sensitivity to chemotherapeutic agents [40], suggesting that they may be an effective therapeutic target [36,37,41]. Analysis of the microRNA expression profile in cancer cells can provide valuable information on the stage of the disease [42,43].

### 3.1. Active Forms of Vitamin D_3_ Affect the Cell Cycle of Human Leukemia and Lymphoma Cells

The antiproliferative effect of calcitriol, through the arrest of cells in the G_0_/G_1_ phase, was observed in colon cancer cells. In these studies, this effect was caused by an increase in the cyclin p21 and p27-dependent kinase inhibitors and a simultaneous decrease in the amount of cyclin A and cyclin E [44]. In pancreatic cancer cells, calcitriol and its analogs also inhibited cells in the G_1_ phase, associated with an increase in p21 and p27 proteins [45]. The p21 protein belongs to the family of cyclin-dependent kinase inhibitors, including the p27 and p57 proteins. The p21 protein is crucial in arresting cell growth following DNA damage. Increasing the amount of this protein leads to an accumulation of cells in the G_1_ phase [46]. Our previous studies also showed that active forms of vitamin D_3_ increased the number of human leukemia and lymphoma cells in G_0_/G_1_ [47]. The use of calcitriol and tacalcitol increased the percentage of cells in the G_0_/G_1_ phase and decreased the number in the S phase in sensitive MV-4-11, Thp-1, and HL-60 leukemia cells. No changes in the cell cycle were observed in the remaining leukemia and lymphoma cells used in the studies. Interestingly, all tested cell lines showed a significantly higher level of p21 protein than normal blood cells. At the same time, calcitriol and tacalcitol did not cause significant changes in the level of p21 protein in all tested cell lines. On the other hand, the amount of p27 protein was comparable in most cell lines to that observed in normal PBMCs. However, two cell lines, Raji and Daudi, expressed significantly higher levels of p27 than control cells. The use of calcitriol and tacalcitol increased the amount of p27 protein in sensitive cells, while in insensitive cells, no changes were observed, or the amount of this molecule was reduced. The results may indicate that the inhibition of sensitive cells in the G_1_/G_0_ phase is due to an increase in p27 protein by calcitriol and tacalcitol. Increasing the expression of p27 by calcitriol and tacalcitol may also be very beneficial in terms of the overall survival in patients with AML. It turns out that high p27 levels extend the OS of AML patients. The use of calcitriol and tacalcitol increases the level of p27, which may be an effective therapeutic tool.

Analysis of the miR-181 molecules and p21 and p27 levels showed that miR-181a was significantly lower in all tested cell lines. In contrast, the level of miR-181b was similar in most of the cell lines tested to that observed in control PBMCs. The use of calcitriol and tacalcitol caused an increase in miR-181a and a decrease in miR-181b in sensitive cells, while in insensitive cells, mainly a decrease in miR-181a and an increase in miR-181b were observed. The obtained results suggest that the p21 protein is not involved in the blockade of sensitive to calcitriol and tacalcitol cells in the G_1_/G_0_ phase, but p27 and its level may depend on the amount of miR-181b molecule. The pioneers who discovered the role of the miR-181b molecule in the regulation of the cell cycle believe that the transition of cells between these phases requires the joint operation of several mechanisms and that interference with only one of the factors regulating this complicated process may have only a moderate impact on its course. They also emphasize that it is unrealistic to expect that the p27 protein is fully controlled only by the miR-181b molecule [25]. Cell cycle control is a very complex process involving many molecular factors, including many miRNA molecules [32]. Interestingly, the induction of the p21 or p27 protein leads to an increase in the expression of the Bak1 protein and the activation of caspase 7 [48]. In the conducted studies, we observed a simultaneous increase in the levels of the Bak1 and p27 proteins. For a full explanation of the mechanism of action of the active forms of vitamin D_3_, it will be rational to determine the activity of caspases.

### 3.2. Population of Aneuploid Cells in Human Leukemia and Lymphoma Cells and Response to Active Forms of Vitamin D_3_

The number of cells with an abnormal number of chromosomes—aneuploids—was also estimated. In leukemia cell lines, the percentage of aneuploids was significantly higher compared to lymphomas. Calcitriol and tacalcitol caused a significant reduction in the rate of aneuploid cells in the sensitive cell lines MV-4-11, Thp-1, and HL-60. The role of aneuploid cells in tumor development has not yet been elucidated. It turns out that the presence of aneuploid cells is responsible for the more aggressive phenotype of cancer [49]. In vitro studies confirm that aneuploid cells have a different growth rate, metabolism, cell cycle kinetics, and size than diploid cells. Numerous studies show that aneuploidy is a hallmark of developing cancer cells. Many mechanisms are responsible for their formation, opening the door for developing new therapeutic strategies against aneuploid cells [50]. Interestingly, studies show that the presence of aneuploid cells increases the sensitivity of tumors to cytostatics, which may also be an excellent target in anti-cancer therapy [51]. Calcitriol and tacalcitol showed the most significant activity against cell lines with the highest percentage of aneuploid cells, suggesting that the sensitivity of human leukemia cells to these substances may depend on the presence of aneuploid cells. On the other hand, knowing that aneuploid cells arise through disturbances in the mechanisms of cell division, it may be suggested that calcitriol and its analogs show selective activity against cells in which these processes are disturbed. However, additional and detailed studies are necessary to explain these phenomena.

### 3.3. Pro-Apoptotic Proteins Bim and Bak1 in the Response of Human Leukemia and Lymphoma Cells to Calcitriol and Tacalcitol

Programmed cell death or apoptosis occurs in response to developmental changes and disease states, and it is required to maintain a state of homeostasis in the body. Cell resistance to apoptosis is one of the hallmarks of cancer cells [52,53]. Proteins of the Bcl-2 family play a key role in regulating apoptosis and modulating death signaling through the internal or mitochondrial pathway [54]. While the Bcl-2 proteins and several close relatives promote cell survival, other family members, such as Bax and Bak, induce apoptosis. For example, the BH3 family members (Bim, Puma, Noxa, Bid, Bad, and Bik) induce apoptosis by activating the Bax and Bak proteins [55]. Interestingly, a low concentration of the Bim protein can activate Bak1. However, to obtain the same level of both proteins, a high concentration (µM) of the Bim protein is necessary [56]. In the conducted studies, the use of calcitriol and tacalcitol resulted in a decrease in the pro-apoptotic protein Bim and an increase in the amount of Bak1 protein. Therefore, examining the status of other pro-apoptotic BH3 proteins can explain the role in regulating Bak1 levels under the influence of calcitriol and tacalcitol.

These results indicate that cells sensitive to calcitriol and its analog may undergo apoptosis by increasing the amount of the Bak1 protein. It is known that miRNA molecules can regulate both tested pro-apoptotic proteins. Studies have shown that the miR-125b molecule regulates Bak1 protein expression. In vivo studies confirmed that inhibition of miR-125b reduced the tumor growth of CML cells in mice. Low miR-125b levels affected the expression of the Bak1 protein, promoting apoptosis and inhibition of cell proliferation [57]. The reduction in Bak1 levels inhibited paclitaxel-induced apoptosis and led to an increase in paclitaxel resistance. In turn, the restoration of Bak1 expression with the miR-125b inhibitor restored the sensitivity of these cells. [58]. The conducted studies that indicated that the miR-125b molecule also significantly affects the chemotherapy of ovarian cancer cells. The increase in the expression of this molecule contributes to cisplatin resistance by inhibiting the amount of the Bak1 [59]. The pro-apoptotic protein Bim is regulated by the miR-32 molecule and miR-181a [60]. This molecule is crucial in the apoptosis of cancer cells of epithelial origin and the response to paclitaxel chemotherapy. Lowering the amount of Bim protein by the miR-32 molecule may contribute to the resistance of cancer cells to the induction of apoptosis in the tumor environment [61,62]. Interestingly, the level of Bim protein in all leukemia cell lines was higher than that observed in normal cells. However, this level in lymphoma cells was similar to the level present in control cells. Furthermore, the use of calcitriol and tacalcitol resulted in a decrease in Bim and an increase in the amount of the miR-32 molecule, which suggests that the amount of this pro-apoptotic protein may be regulated by the miR-32 molecule. Moreover, very interesting observations were made based on the analysis of the survival of patients with AML and the level of expression of miR-32, Bak1, and Bim. The high level of Bak1 and Bim expression correlates with a shorter overall survival. On the other hand, a high level of miR-32 is a good prognostic marker and extends the survival rate. Moreover, the use of calcitriol and tacalcitol causes an increase in miR-32 and a simultaneous reduction if Bim in sensitive cells. These observations suggest that calcitriol and tacalcitol may be useful treatment for regulating gene expression and increase the overall survival prognostic factor in AML patients AML. On the other hand, the level of the pro-apoptotic Bak1 protein in cells not treated with calcitriol and tacalcitol was similar to the control cells (except for Raji and Daudi cells). There were also no differences in the basal level of this protein between sensitive and insensitive cells. The use of calcitriol and tacalcitol increased the amount of Bak1 and miR-125b proteins, indicating that this molecule probably does not affect the level of Bak1 protein in human leukemia and lymphoma cells.

### 3.4. miR-125b and Proteins Involved in Cell Proliferation and Survival: NFκB and p53

#### NFkB

One of the critical factors that initiate, or block, apoptosis is the transcription factor NF-κB, which can be affected by p53. In the conducted studies, no relationship was observed between the basal amount of the NF-κB protein and the sensitivity of cells to calcitriol and tacalcitol. Studies indicate that blocking the NF-κB activity plays a crucial role in p53-induced cell death. It turns out that inhibition of NF-κB in cancer cells with an unmutated form of p53 leads to a decrease in response to cytostatics [63]. The nuclear factor-kappa B belongs to the family of transcription proteins, responsible for regulating the inflammatory process, immune response and proliferation, and apoptosis. Constitutive activation of the NF-κB pathway is often present in many types of cancer and chronic inflammatory diseases, such as multiple sclerosis, enteritis, rheumatoid arthritis, or asthma [64]. The regulation of cellular metabolism by NF-κB depends on the protein p53. Many mutations, including those involving p53, contribute to the activation of NF-κB in cancer cells. The p53 protein is antagonistic to the factor NF-κB, which may inhibit cancer development. A mutation in the p53 blocks the NF-κB pathway [65]. In addition, this factor can interfere with p53 transcriptional activity by lowering the p53 levels and inhibiting apoptosis by increasing the anti-apoptotic proteins [66].

In the conducted studies, it was observed that the level of NF-κB factor is reduced, in both sensitive and insensitive cells, after using calcitriol. The miR-125b molecule is known to be involved in regulating the expression of NF-κB [67]. It turns out that the level of miR-181b can also indirectly affect the amount of this transcription factor by regulating the CYDL protein. This protein is a tumor suppressor that lowers the NF-κB levels. It turns out that an increase in the amount of miR-181b causes a slowdown in CYDL production and an increase in the amount of NF-κB [68]. Interestingly, the transcription factor NF-κB can induce apoptosis and increase the amount of p21 protein in Ewing sarcoma cells through a pathway independent of the p53 protein. Based on the conducted research, it was found that an increase in the amount of the p21 protein protected cancer cells from apoptosis induced by TNF-α [69]. In other studies, it has also been found that monocyte apoptosis induced by TNF-α can be inhibited by an increase in p21 via NF-κB [70]. In these studies, a decrease in the level of transcription factor NF-κB was related to an increase in the amount of the miR-125b molecule, which may indicate that this molecule may affect the level of the NF-κB protein. We did not observe significant changes in the p21 and p53 protein levels (Appendix A). In cells sensitive to calcitriol and tacalcitol, a decrease in the amount of miR-181b was observed, and in insensitive cells, an increase in miR-181b, which may indicate that this molecule is not involved in the regulation of the NF-κB levels in human leukemia and lymphoma cells by calcitriol or its analog. Interestingly, calcitriol can block the NF-κB. Studies showed that VDR interacts with IKKβ, which is necessary for NF-κB activation [71,72]. On the other hand, cells without VDR are characterized by an increased level of NF-κB [73]. The simultaneous increase in the VDR and miR-125b levels is responsible for the lowering of the NF-kB level, but further analyses are required. Interestingly, high levels of NFkB are associated with a reduced life expectancy in AML patients. The action of calcitriol and tacalcitol in the human leukemias and lymphomas is presented in the Figure 12.

### 3.5. miR-125b and miR-27b in the Regulation of Classic Vitamin D Receptor and CYP24A1

Our recent studies showed that the level of two receptors, the classic vitamin D receptor (VDR) and 1,25D_3_-MARRS, increased at the same time under the influence of active forms of vitamin D_3_. This effect was observed after calcitriol and tacalcitol use in three cell lines, MV-4-11, Thp-1, and HL-60. In the case of the remaining cells used in the studies, an increase in one of the tested receptors or a decrease in the level of these molecules was observed, and a simultaneous lack of sensitivity to calcitriol and tacalcitol [74]. It turns out that the response to calcitriol and tacalcitol via the vitamin D-VDR receptor may depend on the level of miRNA molecules that regulate the level of this protein. Research shows that two miRNAs, miR-27b and miR-125b, can control the amount of the vitamin D receptor—VDR [75]. The analysis of the level of these molecules showed that almost all leukemia and lymphoma cells selected for the study had significantly lower levels of miR-27b and miR-125b. The higher miR-125b characterized only KG-1 and U2932 compared to normal cells. In turn, the use of calcitriol and tacalcitol resulted in a significant increase in the amount of these molecules in all cell lines selected for research. The results indicate that the increase in miR-27b and miR-125b molecules did not reduce the level in cells sensitive to calcitriol and tacalcitol. Research indicates that the expression of miR-27b and miR-125b can be activated by the action of VDR [20,21]. However, the detailed molecular mechanism has not been known so far. Therefore, a thorough understanding of miR-27b and miR-125b, and vitamin D_3_, should be of interest to future research. Interestingly, high levels of miR-27b and miR-125b extend survival in AML patients. The use of calcitriol and tacalcitol increased the expression of these molecules. It seems that the use of these substances in patient treatment could significantly extend the survival rate. Therefore, further work is needed on the precise understanding of their molecular interactions. Significantly lower levels of CYP24A1 mRNA characterized human leukemia and lymphoma cells selected for the study compared to normal cells. In contrast, the level of CYP24A1 protein was significantly higher in all neoplastic cells than in control cells. No differences in the level of this enzyme were observed between sensitive and insensitive cells, which may suggest that the basic level of this enzyme does not determine the sensitivity of leukemia and lymphoma cells to calcitriol and tacalcitol. The use of calcitriol and tacalcitol increased the amount of CYP24A1 mRNA in all cancer cell lines used in the study. In turn, the protein level analysis showed that the increase in the level of this molecule was observed only in cells sensitive to calcitriol and tacalcitol. In contrast, in non-sensitive cells, a decrease in the level of the CYP24A1 protein was observed. A vitamin D response element in the gene’s promoter region encodes the CYP24A1 enzyme. Accumulating evidence suggests that high levels of CYP24A1 affect the biological activity of calcitriol in cancer cells. The combined use of calcitriol and CYP24A1 inhibitors increases the antiproliferative activity of calcitriol and its analogs [76]. The amount of CYP24A1 protein may depend on the level of the miR-125b molecule [77]. In the conducted studies, an increase in the level of the miR-125b molecule was observed in all tested cell lines. In cells sensitive to calcitriol and tacalcitol, an increase in the level of the CYP24A1 protein was observed. In contrast, in insensitive cells, a decrease in the amount of this enzyme was mainly observed. In cells sensitive to calcitriol and tacalcitol, along with an increase in the CYP24A1 protein, an increase in the miR-125b molecule was noticed, suggesting that this molecule may not affect the level of this protein in these cells. In contrast, in cells insensitive to calcitriol and tacalcitol, the level of CYP24A1 may be decreased by the miR-125b molecule. Interestingly, only in cells sensitive to the action of calcitriol and tacalcitol, did we observe an increase in VDR and CYP24A1 expression [74]. VDR is known for activating the CYP24A1 promoter region and initiating CYP24 transcription. It is also known that calcitriol actively induces CYP24 splicing [78]. Further research is required to identify the mechanisms responsible for this differential response of sensitive and insensitive cells to vitamin D_3_ derivatives.

## 4. Materials and Methods

### 4.1. Human Leukemia and Lymphoma Cell Lines

The following human cancer cell lines were used: leukemia: KG-1, K562, HL-60, MV-4-11, Thp-1; lymphoma–Jurkat, U2932, Daudi, and Raji. All cells were derived from the Hirszfeld Institute of Immunology and Experimental Therapy repository, Polish Academy of Sciences in Wroclaw, Poland. All tested cell lines were cultured in RPMI 1640w/GLUTAMAX-I media (Gibco, Gaithersburg, MD, USA) containing 10% fetal bovine serum (Thermo Fisher Scientific, Waltham, MA, USA), streptomycin (0,1 mg/mL), and penicillin (100 U/mL) (Polfa Tarchomin, Warsaw, Poland). In addition, media for Thp-1 were supplemented with 0.05 mM β-mercaptoethanol, MV-4-11, and HL-60 with sodium pyrogroniate 1 mM and HL-60 additionally with 3.5 g/L glucose (all Sigma-Aldrich, Steinheim, Germany). Cells were incubated at 37 °C with a 5% CO_2_ in a humid atmosphere incubator (NuAire, Plymouth, MN, USA).

### 4.2. Human Normal Blood Cells

Samples of peripheral blood from 10 healthy people who consciously and voluntarily submitted them for testing and signed individual statements were collected after obtaining the consent of the Bioethics Committee at the Medical University of Wroclaw, No. 71/2017. First, whole blood was collected into test tubes with EDTA anticoagulant by specialized personnel of the Diagnostics laboratory at the Institute of Immunology and Experimental Therapy. According to the manufacturer’s instructions, blood mononuclear cells were then isolated using the Ficoll-Paque PLUS kit (GE Healthcare, Amersham, UK).

### 4.3. Calcitriol and Tacalcitol

The active forms of vitamin D, calcitriol (1,25-(OH)_2_D_3_) and tacalcitol (PRI-2191, 1,24-(OH)_2_D_3_, were used. Compounds were synthesized at the Pharmaceutical Research Institute (PRI, Warsaw, Poland). Stock solutions of calcitriol and tacalcitol were obtained by dissolving 50 µg lyophilizate in 99.8% ethanol (Avantor, Gliwice, Poland), obtaining a 10^–4^ nM solution, and was stored at −20 °C.

### 4.4. Real-Time PCR

Cells suspended in the media were centrifuged for 5 min at 300× *g* and washed with PBS. The cell pellet was resuspended in 700 µL of TRIzol (Invitrogen, Carlsbad, CA, USA). Then 0.2 mL of chloroform (Avantor, Gliwice, Poland) was added and mixed for 15 s. Samples were incubated for 5 min at room temperature and centrifuged for 15 min at 12,000× *g* at 4 °C. The aqueous phase was transferred to new tubes, and then 0.5 mL of isopropanol (Avantor, Gliwice, Poland) was added. Samples were incubated for 5 min at room temperature and centrifuged for 15 min at 12,000× *g* at 4 °C. The supernatant was removed, and the RNA pellet was resuspended in 1 mL of 75% ethanol with DEPC. Samples were centrifuged for 5 min 7500× *g* at 4 °C. The supernatant was removed, and the samples were air-dried for 5 min at room temperature. Each sample was dissolved in 30 μL of molecular biology-grade water (Sigma-Aldrich, Steinheim, Germany) and left on ice until completely dissolved. The RNA concentration was measured using a NanoDrop 2000 spectrophotometer (ThermoFisher Scientific, Waltham, MA, USA).

A total of 1 µg of RNA was used for purification. The following mixture was prepared: 1 μg RNA, 1 μL RNAz inhibitors, 1.5 μL DNAseI buffer, 1 μL DNAseI enzyme (all reagents from ThermoFisher Scientific, Waltham, MA, USA), made up to 15 μL with molecular biology water (Sigma-Aldrich, Steinheim, Germany). DNA digestion was carried out in a Veriti thermal cycler (Applied Biosystems, Foster, CA, USA) at 37 °C for 15 min. After incubation, 2 µL EDTA (ThermoFisher Scientific, Waltham, MA, USA) was added to the samples and incubated for 10 min at 65 °C to inactivate the enzyme. Electrophoresis was performed on both samples after and before cleaning.

For mRNA, 15 µL of the mixture after purification, 4 µL of a 5× concentrated buffer for reverse transcriptase, and 1 µL of reverse transcriptase enzyme from the iScript kit (BioRad, Hercules, CA, USA) were added. The reverse transcription process was carried out in a Veriti thermal cycler under the following conditions: 5 min at 25 °C, 30 min at 42 °C, and 5 min at 85 °C. Samples were stored at −20 °C. For miRNA, 1 µg of RNA was used for reverse transcription. The reaction mixture was then prepared using the TaqMan MicroRNA Reverse Transcription Kit (ThermoFisher Scientific, Waltham, MA, USA). The following reverse transcription microRNA probes were used: TaqMan MicroRNA Assay RT hsa-miR-27b, TaqMan MicroRNA Assay RT has-miR-32-5p, TaqMan MicroRNA Assay RT hsa-miR-125b-5p, TaqMan MicroRNA Assay RT hsa-miR-181a-5p, TaqMan MicroRNA Assay RT hsa-miR-181b-3p. The reverse transcription reaction was performed in a Veriti thermocycler under the following conditions: 30 min at 16 °C, 30 min at 42 °C, and 5 min at 85 °C. After reverse transcription, the samples were stored at −20 °C. To analyze vitamin D receptor (VDR) (Hs01045840_m1), 1.25D3-MARRS (Hs00607126_m1), and CYP24A1 (Hs00167999_m1) expression, 50 ng of cDNA was used. The final reaction volume was 10 µL, and TaqMan Universal PCR Master Mix (Applied Biosystems, Foster, CA, USA) and TaqMan probes were used. 50 ng of cDNA per reaction was used to examine the expression of selected microRNA molecules. The final volume of the reaction mixture was 10 µL. The template also contained TaqMan Universal PCR Master Mix, TaqMan Small RNA probes (Applied Biosystems, Foster, CA, USA), and 4.8 µL of reverse transcription product in addition to the template. The following probes were used to perform the Real-Time PCR reaction: TaqMan MicroRNA Assay TM hsa-miR-27b, TaqMan MicroRNA Assay TM hsa-miR-32-5p, TaqMan MicroRNA Assay TM hsa-miR-125b-5p, TaqMan MicroRNA Assay TM hsa-miR-181a-5p, and TaqMan MicroRNA Assay TM hsa-miR-181b-3p. The reaction was run under the following conditions: 95 °C for 10 min, 40 cycles consisting of denaturation at 95 °C for 15 sec, and annealing at 60 ° C for 1 min in a Real-Time PCR Via 7 Reaction Machine (Applied Biosystems, Foster, CA, USA). Expression of the tested genes was estimated using the comparative method ∆∆Ct. Data were analyzed using Data Assist v. 3.01 (Applied Biosystems, Foster, CA, USA). At least three independent RNA isolations were performed for all cell lines used in the study.

### 4.5. ELISA Tests

Commercial ELISA assays (Elabscience, Houston, TX, USA) were used to determine the CYP24A1, p53, NFκB, Bak1, Bim, p21, and p27 proteins. The manufacturer’s instructions were followed in all steps. Standard curves were prepared by diluting the standard for each protein to be assayed in 1 mL Reference Standard & Sample Dilution. The standard dilutions were prepared as follows: CYP24A1 (10; 5; 2.5; 1.25; 0.63; 0.31; 0.16; 0 ng/mL), p53 (5000; 2500; 1250; 625; 312.5; 156.25; 78.13; 0 pg/mL), NFκB (4000; 2000; 1000; 500; 250; 125; 62.5; 0 pg/mL), Bak1 (10; 5; 2.5; 1.25; 0.63; 0.31; 0.16; 0 ng/mL), Bim (20; 10; 5; 2.5; 1.25; 0.625; 0.313; 0 ng/mL), p21 (1000; 500; 250; 125; 62.5; 31.25; 15.63; 0 pg/mL), p27 (20; 10; 5; 2.5; 1.25; 0.625; 0.313; 0 ng/mL). In total, 100 ng of each sample were diluted 25× in Reference Standard & Sample Dilution. Then 100 μL of the sample and the standard were added to the wells of a 96-well coated plate. Samples were incubated for 90 min at 37 °C. The solution in the well was then carefully removed, and 100 µL of biotinylated Biotinylated Detection Ab was immediately added and incubated for 1 h at 37 °C. After incubation, the wells were washed three times with 350 μL Wash Buffer, and 100 µL of the HRP conjugate solution was added to each well and incubated for 30 min at 37 °C. Then the plates were washed 5 times with Wash Buffer. In total, 90 µL of Substrat Solution were added, and plates were incubated in the dark for 15 min at 37 °C. After a time, 50 µL of Stop Solution was added. The color of the solution turned from blue to yellow. Spectrophotometric measurement was performed at a wavelength of 450 nm using a Synergy H4 Hybrid Multi-Mode Microplate Reader (BioTek Instruments, Inc., Winooski VT, USA). Standard curves were plotted using the Curve Expert software (Hyams, D.G., CurveExpert software, http://www.curveexpert.net, 2010 (accessed on 10 August 2017)).

### 4.6. Cell Cycle Analysis and Aneuploids Population

Cells (10^5^ cells/well) were plated in 24-well plates (Corning Incorporated, Corning, NY, USA) in 2 mL of media. After 24 h, 10 nM of calcitriol and tacalcitol were added. Cells were harvested at 24, 48, 72, 96, and 120 h in 15 mL tubes (Corning Incorporated, Corning, NY, USA), centrifuged at 400× *g* at 4 °C for 10 min. The supernatant was removed, and cells were washed with PBS and counted. One million cells were resuspended in 0.7 mL of 70% cold ethanol and stored at −20 ° C for a minimum of 24 h. On the day of the cell cycle analysis, the 70% ethanol was removed by centrifuging the samples at 400× *g* at 4 °C for 10 min. Then the cell pellet was washed by adding 0.5 mL PBS and resuspended in an 8 µg/mL RNAse solution (ThermoFisher Scientific, Waltham, MA, USA), transferred to cytometer tubes (Corning Incorporated, Corning, NY, USA), and incubated for 1 h at 37 °C. After incubation, 50 µL of 0.1 mg/mL propidium iodide (ThermoFisher Scientific, Waltham, MA, USA) was added to the cells. After 30 min incubation at room temperature, samples were analyzed using a BD LSRFortessa flow cytometer using FACS Diva software (Becton Dickinson, San Jose, CA, USA). The percentage of cells in particular phases of the cell cycle was determined using ModFit LT software (Verity Software House, Topsham, ME, USA).

### 4.7. Statistical Analysis

Statistical analysis of the obtained results was performed using the Statistica 7.1 program (StatSoft Inc., Tulsa, USA). Before selecting the test, the normality of the distribution was analyzed using Levene’s test. Next, data showing a normal distribution were analyzed using one-way ANOVA. Finally, non-parametric data were analyzed with the Kruskal–Wallis test. Results were statistically significant when *p* < 0.05. The charts were prepared using Excel (Microsoft Office) and GraphPad Prism version 6.04 for Windows (GraphPad Software, La Jolla, CA, USA).

### 4.8. Overall Survival Analysis

TCGA survival data was linked to VDR, Bak1, Bim, p21, p27, p53, miR-27b, miR-32, miR-125b, miR-181a, and miR-181b expression in patients with acute myeloid leukemia by downloading data from http://www.oncolnc.org/ (accessed on 18 April 2022). Survival data up to 3000 days was used for the survival analysis. Kaplan–Meier curves and log-rank tests were performed using the survival package [79].

## 5. Conclusions

The studies indicate that the antitumor activity of calcitriol and tacalcitol results from the interaction of several miRNA molecules and the level of the targeted protein. Active forms of vitamin D_3_ caused an increase in the p27 protein that blocked cells in the G_0_/G_1_ phase of the cell cycle. At the same time, there was an increase in the pro-apoptotic protein Bak1. It correlated with the decreased level of miR-181b regulating p27. Interestingly, the use of calcitriol and tacalcitol resulted in an increase in miR-27b and miR-125b as well as CYP24A1, with a simultaneous reduction in NF-kB. These two miRNAs’ (miR-27b and miR-125b) roles should be studied further to understand the molecular mechanism of vitamin D_3_ action.

## Figures and Tables

**Figure 1 ijms-23-05019-f001:**
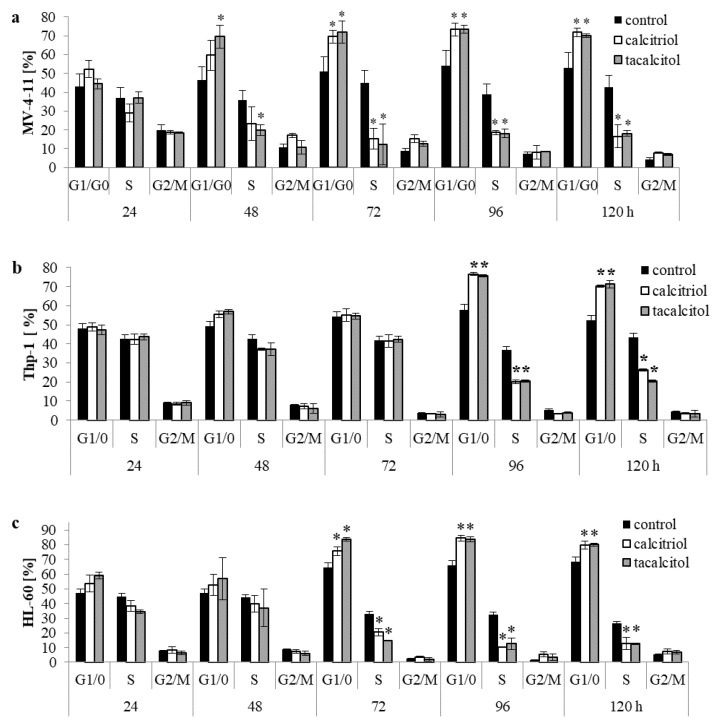
Cell cycle analysis of leukemia cells: acute biphenotypic myelomonocytic leukemia MV-4-11 (**a**), acute monocytic leukemia Thp-1 (**b**), and acute promyelocytic leukemia HL-60 (**c**) after the use of calcitriol and tacalcitol. The cell cycle was analyzed 24, 48, 72, 96, and 120 h after incubation with 10 nM calcitriol and tacalcitol. The graph shows the mean ± standard deviation. *—statistical significance (*p* < 0.05) compared to the control.

**Figure 2 ijms-23-05019-f002:**
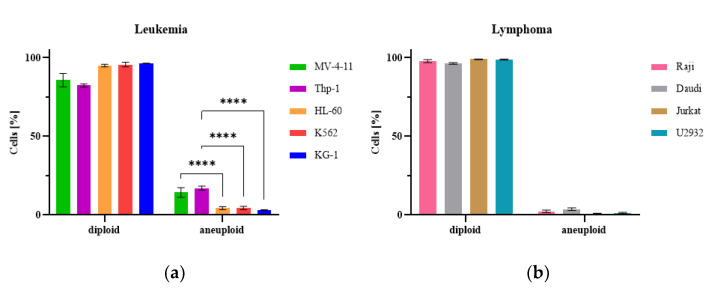
Diploids and aneuploids in human leukemia (**a**) and lymphoma (**b**) cells. The highest percentage of cells with abnormal chromosome numbers (aneuploids) was observed in the two sensitive to calcitriol and tacalcitol leukemia cell lines: MV-4-11 and Thp-1. The graph shows the mean ± standard deviation (n = 3). **** *p* < 0.0001.

**Figure 3 ijms-23-05019-f003:**
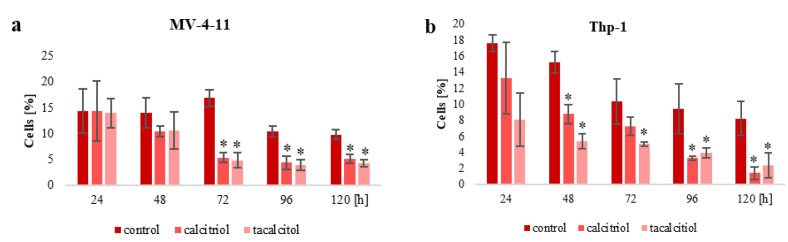
The population of aneuploids in leukemia cells after calcitriol and tacalcitol use. Active forms of vitamin D_3_ reduced the percentage of aneuploids in MV-4-11 (**a**), Thp-1 (**b**), and HL-60 (**c**). The graph shows the mean ± standard deviation. *—statistical significance (*p* < 0.05) compared to the corresponding time point.

**Figure 4 ijms-23-05019-f004:**
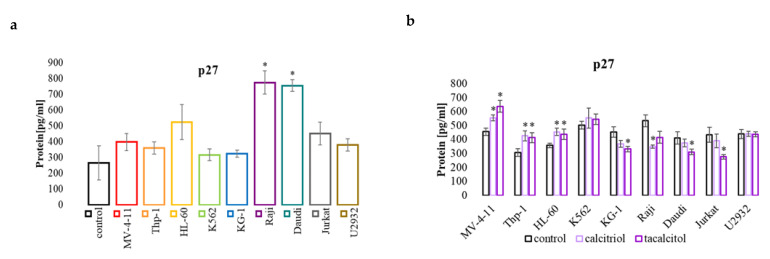
The p27 (**a**) protein (ELISA test) in human leukemia and lymphoma cells compared to the control: (**a**) PBMCs; (**b**) untreated cells of the appropriate cell line. miR-181b (**c**) in human leukemia and lymphoma cells compared to the control: (**c**) PBMCs; (**d**) untreated cells of the appropriate cell line. The relative quantification (RQ) was calculated using as control: (**a**) and (**c**) PBMCs; (**d**) and untreated cells of the appropriate cell line. *—statistical significance (*p* < 0.05) compared to the control.

**Figure 5 ijms-23-05019-f005:**
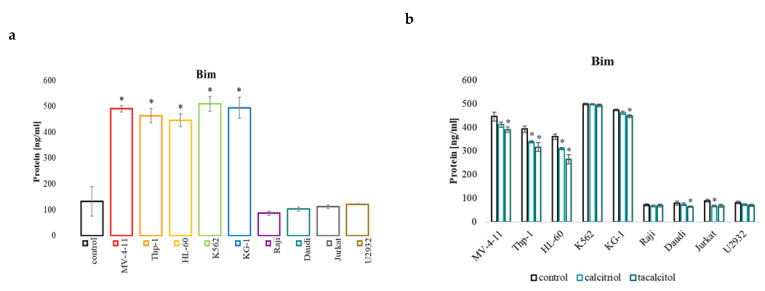
Pro-apoptotic protein Bim (ELISA test) level (**a**), and miR-32 molecule (**c**) in human leukemia and lymphoma compared to the control: (**a**) and (**c**) PBMCs; (**b**,**d**) untreated cells of the appropriate cell line. The relative quantification (RQ) was calculated using as the control: (**a**,**c**) PBMCs; (**d**) untreated cells of the appropriate cell line. The Bim (**b**) and miR-32 (**d**) after use of calcitriol and tacalcitol. *—statistical significance (*p* < 0.05) compared to the control.

**Figure 6 ijms-23-05019-f006:**
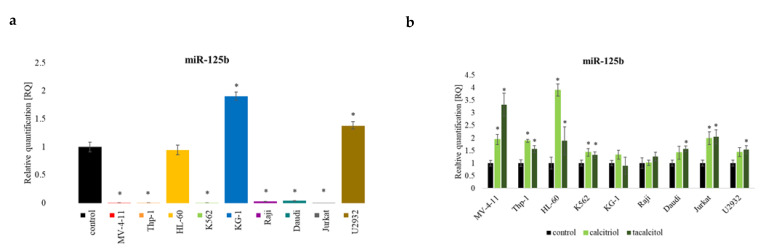
The miR-125b in human leukemia and lymphoma cells compared to the control: (**a**) PBMCs; (**b**) untreated cells of the appropriate cell line. The relative quantification (RQ) was calculated using as control: (**a**) PBMCs; (**b**) untreated cells of the appropriate cell line. The miR-125b in tested cell lines (**a**), and after incubation with calcitriol and tacalcitol (**b**). The graph shows the mean ± standard deviation. *—statistical significance (*p* < 0.05).

**Figure 7 ijms-23-05019-f007:**
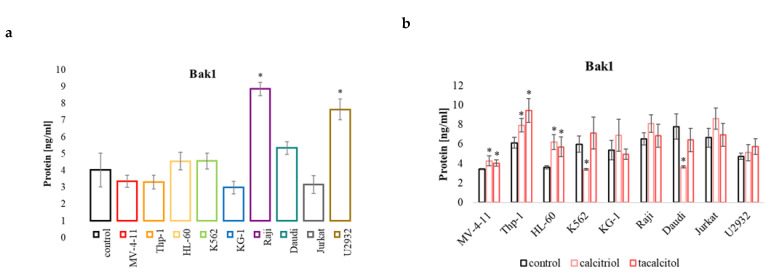
Bak1 in human leukemia and lymphoma cells compared to the control: (**a**) PBMCs; (**b**) untreated cells of the appropriate cell line. The relative quantification (RQ) was calculated using as the control: (**a**) PBMCs. The Bak1 level in the tested cell lines (**a**), and after incubation with calcitriol and tacalcitol (**b**). The graph shows the mean ± standard deviation. *—statistical significance (*p* < 0.05).

**Figure 8 ijms-23-05019-f008:**
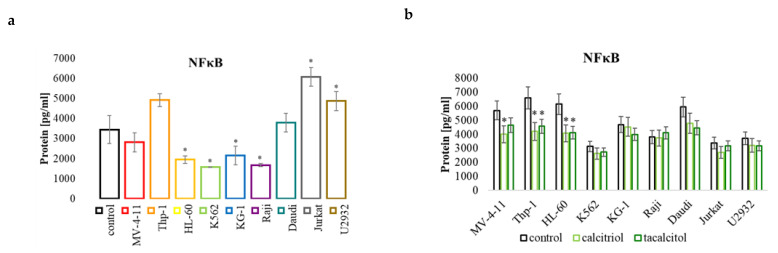
NFκB in human leukemia and lymphoma cells compared to the control: (**a**) PBMCs; (**b**) untreated cells of the appropriate cell line. The relative quantification (RQ) was calculated using as control: (**a**) PBMCs. NFκB in the tested cell lines (**a**), and after incubation with calcitriol and tacalcitol (**b**). The graph shows the mean ± standard deviation. *—statistical significance (*p* < 0.05).

**Figure 9 ijms-23-05019-f009:**
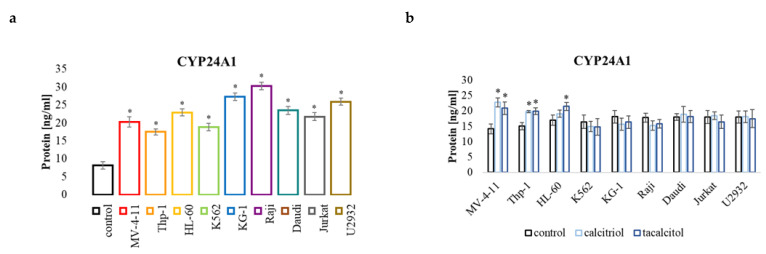
CYP24A1 in human leukemia and lymphoma cells compared to the control: (**a**) PBMCs; (**b**) untreated cells of the appropriate cell line. The relative quantification (RQ) was calculated using as control: (**a**) PBMCs. CYP24A1 in the tested cell lines (**a**), and after incubation with calcitriol and tacalcitol (**b**). The graph shows the mean ± standard deviation. *—statistical significance (*p* < 0.05).

**Figure 10 ijms-23-05019-f010:**
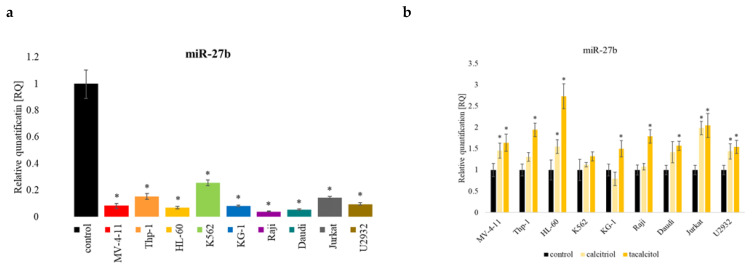
miR-27b in human leukemia and lymphoma cells compared the control: (**a**) PBMCs; (**b**) untreated cells of the appropriate cell line. The relative quantification (RQ) was calculated using as control: (**a**) PBMCs; (**b**) untreated cells of the appropriate cell line. The miR-27b in the tested cell lines (**a**), and after incubation with calcitriol and tacalcitol (**b**). The graph shows the mean ± standard deviation. *—statistical significance (*p* < 0.05).

**Figure 11 ijms-23-05019-f011:**
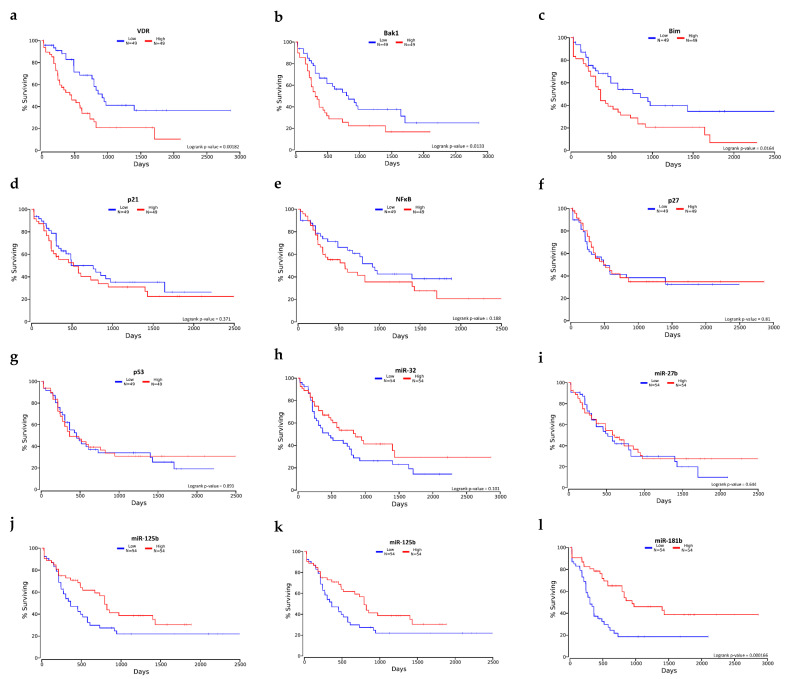
Kaplan–Meier overall survival (OS) for AML patients with high (red) and low (blue) expression of tested mRNA and miRNA. The VDR (**a**), Bak1 (**b**), Bim (**c**), p21 (**d**), and NFκB (**e**) high expression showed shorter overall survival (OS) compared to the low expression. On the other hand, high expression of p27 (**f**), p53 (**g**), miR-32 (**h**), miR-27b (**i**), miR-125b (**j**), miR-181a (**k**), and miR-181b (**l**) showed longer overall survival compared to low expression.

**Figure 12 ijms-23-05019-f012:**
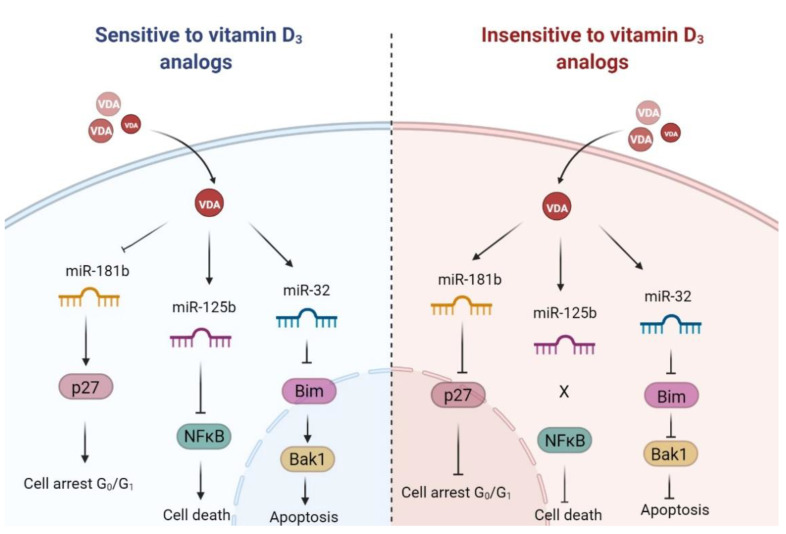
Vitamin D_3_ analogs (VDA) biological activity in leukemia and lymphoma cells. The effect on miRNA and protein expression in sensitive and insensitive to VDAs leukemia and lymphoma cells.

## Data Availability

The data presented in this study are available in this article (and Appendix A).

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
