# Peer review of "Micro-RNAs in Response to Active Forms of Vitamin D3 in Human Leukemia and Lymphoma Cells"

_ijms, 2022, doi:10.3390/ijms23095019_

Round 1

Reviewer 1 Report

In this work, the authors have investigated the role of selected miRNAs and proteins in response to active forms of vitamin D3, calcitriol, and tacalcitol, in a panel of leukemia and lymphoma cell lines. At its present state, the work lacks a rationale, and the findings are not significant enough to be considered for publication, without additional information.

MAJOR COMMENTS:

  1. What is the relevance of studying calcitriol, and tacalcitol treatment in leukemia, and lymphoma?
  2. Are the leukemia, and lymphoma cell lines exhibiting morphological changes, and other features associated with cell differentiation, in response to calcitriol, and tacalcitol treatment?
  3. Can the authors analyze publicly available datasets, where leukemia cells, and/or lymphoma cells have been treated with calcitriol, and tacalcitol, to find differentially expressed genes, and see if the genes they chose for their analysis are the most significantly altered genes in those conditions?
  4. Can the authors do in silico analysis of miRNA target prediction, to find out which miRNA-mRNA pairs should be the most relevant ones for studying, in the context of calcitriol, and tacalcitol treated leukemia/lymphoma cell lines?
  5. How can the authors ensure that their observations of the effect of calcitriol, and tacalcitol treatment on the selected miRNAs, and mRNAs are specific to those molecules only?
  6. Even if these miRNAs, or mRNAs are altered in expression, in response to calcitriol, and tacalcitol treatment, what physiological relevance do they serve? Can the authors analyze publicly available transcriptional datasets of patient samples, like the TCGA, and/or other data sets, to check if any of these miRNAs/mRNAs exhibit prognostic relevance?

Author Response

Dear Reviewer, 

thank you for reviewing our article.

Please see the document with answers to all of your questions. 

Best regards,

Justyna Gleba

Reviewer 2 Report

Although the paper could have been much more substantial by confirming cell line results in experiments in primary tumor cells,  the experiments contribute to the understanding of the well known association of vitamin D deficiency and hematologic malignancies like AML and lymphoma. Only minor spellcheck is required (mainly singular-plural disagreements) which can be  performed at proofreading.  

Author Response

Dear Reviewer, 

thank you for reviewing our article. All spelling changes were made. 

Best regards, 

Justyna Gleba

Round 2

Reviewer 1 Report

The authors have addressed comments satisfactorily. I would additionally suggest them to analyze TCGA data sets for leukemia, and lymphoma, and establish the prognostic relevance of the mRNAs, and miRNAs in these cancers.

Author Response

Dear Reviewer, 

thank you for your work and great suggestions. 

Please see the updated manuscript. 

Best regards, 

Justyna Gleba
